# Partial Endothelial Nitric Oxide Synthase Deficiency Exacerbates Cognitive Deficit and Amyloid Pathology in the APPswe/PS1ΔE9 Mouse Model of Alzheimer’s Disease

**DOI:** 10.3390/ijms23137316

**Published:** 2022-06-30

**Authors:** Sara Ahmed, Yu Jing, Bruce G. Mockett, Hu Zhang, Wickliffe C. Abraham, Ping Liu

**Affiliations:** 1Department of Anatomy, School of Biomedical Sciences, Brain Health Research Centre, University of Otago, Dunedin 9054, New Zealand; ahmsa026@student.otago.ac.nz (S.A.); rena.jing@otago.ac.nz (Y.J.); 2Department of Psychology, Brain Health Research Centre, University of Otago, Dunedin 9054, New Zealand; bruce.mockett@otago.ac.nz (B.G.M.); cliff.abraham@otago.ac.nz (W.C.A.); 3School of Pharmacy, Brain Health Research Centre, University of Otago, Dunedin 9054, New Zealand; hu.zhang@otago.ac.nz

**Keywords:** endothelial nitric oxide synthase, Alzheimer’s disease, cognitive function, amyloid beta, microglia, APP/PS1 mice

## Abstract

Increasing evidence implicates endothelial dysfunction in the pathogenesis of Alzheimer’s disease (AD). Nitric oxide (NO) derived from endothelial NO synthase (eNOS) is essential in maintaining cerebrovascular function and can modulate the production and clearance of amyloid beta (Aβ). APPswe/PSdE1 (APP/PS1) mice display age-related Aβ accumulation and memory deficits. In order to make the model more clinically relevant with an element of endothelial dysfunction, we generated APP/PS1/eNOS^+/−^ mice by crossing complete eNOS deficient (eNOS^−/−^) mice and APP/PS1 mice. APP/PS1/eNOS^+/−^ mice at 8 months of age displayed a more severe spatial working memory deficit relative to age-matched APP/PS1 mice. Moreover, immunohistochemistry and immunoblotting revealed significantly increased Aβ plaque load in the brains of APP/PS1/eNOS^+/−^ mice, concomitant with upregulated BACE-1 (hence increased Aβ production), downregulated insulin-degrading enzyme (hence reduced Aβ clearance) and increased immunoreactivity and expression of microglia. The present study, for the first time, demonstrated that partial eNOS deficiency exacerbated behavioral dysfunction, Aβ brain deposition, and microglial pathology in APP/PS1 mice, further implicating endothelial dysfunction in the pathogenesis of AD. The present findings also provide the scientific basis for developing preventive and/or therapeutic strategies by targeting endothelial dysfunction.

## 1. Introduction

Alzheimer’s disease (AD) is a neurodegenerative disorder and the most common form of dementia in the aged. It is characterized by the neuropathological hallmarks of amyloid plaques and neurofibrillary tangles (NFTs) formed by amyloid beta (Aβ) peptides and hyperphosphorylated tau protein, respectively [1]. Regarding the primary pathophysiological event in AD, a commonly held view centers on the amyloid cascade hypothesis [2]. This hypothesis postulates that AD begins with amyloid precursor protein (APP) derived toxic Aβ species, which lead to the formation of extracellular Aβ plaques and intraneuronal tau neurofibrillary tangles, along with neurodegeneration, inflammation, and vascular damage, ultimately resulting in memory loss. While the amyloid cascade hypothesis has been leading the field for decades, the causality of Aβ in AD (the late-onset form in particular) has been increasingly challenged [3]. There is accumulating evidence suggesting that cerebrovascular dysfunction during advanced aging triggers the neurodegenerative processes in AD [3,4,5].

Nitric oxide (NO) is a gaseous signaling molecule produced by nitric oxide synthase (NOS) from L-arginine [6]. NO can be produced by the vessel endothelial cells via endothelial NOS (eNOS). In the brain, eNOS-derived NO causes the local vascular smooth muscles to relax and hence plays a critical role in regulating cerebral blood flow and regional vascular tone [7]. Considering the brain’s critical dependence on a finely regulated blood supply, substantial and progressive derangements in eNOS signaling could precipitate cerebrovascular dysfunction and cognitive deficits in AD. Earlier research has reported the reduction of eNOS expression in cerebral vessels in AD brains (virtually no eNOS expression in many small and medium-size leptomeningeal, cortical and white matter vessels), and the inverse correlation with amyloid plaques and NFTs [8,9,10]. Recently, we have observed a dramatic (~95%) reduction in eNOS protein expression in the hippocampus and superior frontal gyrus in the AD cases relative to their age-matched control cases at the mean age of 80 years [11]. There is also a lesser (30–70%) reduction in the 80-year control group when compared to the 60-year control group [11]. Collectively, these studies have demonstrated the presence of eNOS deficiency in the brains of AD patients and individuals with advanced age. Such age-related eNOS deficiency may be an important early event contributing to later neurodegeneration in AD [3,4,5]. 

A number of animal studies have explored the link between eNOS deficiency and AD pathologies. Mice with complete or partial eNOS deficiency (eNOS^−/−^ or eNOS^+/−^, respectively) display increased levels of APP, β-site APP cleaving enzyme 1 (BACE1) and Aβ, amyloid angiopathy, and altered neuronal p25 (an aberrant activator of the tau kinase, cyclin-dependent kinase 5 (Cdk5)) in the brain, as well as memory deficits [12,13,14,15,16]. Intriguingly, eNOS-derived NO can directly modulate the production of Aβ and protect against increases in Aβ [12,17]. These findings suggest that loss of eNOS-derived NO contributes to amyloidogenic processing of APP, tau phosphorylation, and cognitive decline. It is of interest to note the alteration of the Aβ clearance pathways in eNOS deficient mice, further indicating a critical role of eNOS in the maintenance of a delicate balance between Aβ production and clearance [13,14,18]. Taken together, the existing evidence from human and animal research strongly implicates eNOS dysfunction in the pathogenesis of AD. 

While the causality of Aβ in AD is heavily debated, Aβ accumulation/deposition in the brain is one of the key neuropathological hallmarks of the disease. Animal models are essential experimental tools to understand the pathogenesis of AD. Transgenic APPswe/PSEN1ΔE9 (APP/PS1) mice are a commonly used model to address the Aβ aspect of AD pathologies, particularly during the early stage of the disease. APP/PS1 mice display progressive Aβ accumulation in the brain and behavioral impairments from 4 months of age, which become more apparent with increased age [19,20]. We have reported no changes in eNOS protein expression in the frontal cortex, hippocampus, parahippocampal region, and cerebellum in APP/PS1 mice at 7 and 13 months of age relative to their age-matched wildtype littermates [20]. Austin and Katusic (2016) developed a new mouse model harboring both APPswe and PSEN1ΔE9 mutations along with complete eNOS deficiency. APP/PS1/eNOS^−/−^ mice at 4–5 months of age displayed markedly higher levels of p25, a higher p25/p35 ratio (indicative of increased Cdk5), and significantly higher Cdk5 activity and increased tau phosphorylation relative to their age-matched APP/PS1 mice [15]. Interestingly, there were no changes in amyloid pathology and neuroinflammation between the two genotype groups at 4–5 months of age. It should be pointed out that APP/PS1/eNOS^−/−^ mice had a high death rate at a younger age due to cardiovascular problems. Moreover, the breeding of APP/PS1/eNOS^−/−^ mice was challenging, as there was only one APP/PS1/eNOS^−/−^ mouse among 22–25 mice born [15]. 

Complete eNOS deficiency in AD brains has not been reported to date. Cardiovascular risk factors of AD, such as hypercholesterolemia, advanced aging, hypertension, and smoking, likely cause partial loss of eNOS expression and function [11]. This has led us to develop a new model by adding partial eNOS deficiency to APP/PS1 mice, a more clinically relevant model for early-stage AD with endothelial dysfunction. The present study, therefore, crossed eNOS^−/−^ and APP/PS1 mice to generate APP/PS1/eNOS^+/−^ mice. We postulated that partial eNOS deficiency would exacerbate the phenotype of APP/PS1 mice, such as more severe behavioral deficits and increased Aβ load in the brain in APP/PS1/eNOS^+/−^ mice. The present study characterized APP/PS1/eNOS^+/−^ mice at 8 months of age to determine the effects of partial eNOS deficiency on cognitive function, Aβ deposition in the brain, and microglial pathology in APP/PS1 mice. Moreover, we further investigated how partial eNOS deficiency affected the enzymes involved in Aβ production and clearance in the brain of APP/PS1 mice, such as an insulin-degrading enzyme (IDE), low-density lipoprotein receptor-related protein-1 (LRP1), and aquaporin-4 (AQP4). 

## 2. Results

### 2.1. General Characterization

In the present study, male APP/PS1 mice and female wildtype littermates were crossed to produce wildtype (WT) and APP/PS1 offspring, whereas male eNOS^−/−^ mice and female APP/PS1 mice were crossed to produce eNOS^+/−^ and APP/PS1/eNOS^+/−^ mice (Figure 1A). The genotype of the offspring was based on the absence (for WT and eNOS^+/−^ mice) or presence (for APP/PS1 and APP/PS1/eNOS^+/−^ mice) of APP and PS1 mutations (Figure 1A). In order to confirm partial eNOS deficiency in eNOS^+/−^ and APP/PS1/eNOS^+/−^ mice, we determined eNOS protein levels in the anterior cortex in all four genotype groups at 8 months of age using western blot (Figure 1B). One-way ANOVA revealed a significant genotype effect (F(3,30) = 20.52, *p* < 0.0001), with a 50% reduction of eNOS protein expression in eNOS^+/−^ and APP/PS1/eNOS^+/−^ mice relative to WT and APP/PS1 mice (all *p* < 0.0001). There was no significant difference in eNOS protein between the WT and APP/PS1 groups.

Animals’ body weights and organ weights (such as brain, liver, spleen, heart, and kidneys) were obtained after completion of behavioral tests. In terms of the body weight (g), there was no significant difference between groups at 8 months of age (WT: 35.73 ± 1.11; eNOS^+/−^: 33.31 ± 1.26; APP/PS1: 33.79 ± 0.68; APP/PS1/eNOS^+/−^: 33.43 ± 1.32; F = 1). When the organ and body weight ratios were analyzed, one-way ANOVA revealed no significant differences across the four genotype groups for all of the variables measured (all F ≤ 2; see Appendix A). 

### 2.2. Behavioural Data

All animals were tested in the open field (day 1), elevated plus maze (day 1), and Y-maze (day 2). Regarding the open field, one-way ANOVA revealed a significant effect of genotype in terms of the path length (F(3,35) = 5.56, *p* = 0.003), with the greater length in the APP/PS1 group relative to the WT (*p* < 0.01) and eNOS^+/−^ (*p* < 0.05) groups (Figure 2A). A significant genotype effect was also observed in the percentage of time spent in the outer zone measurement (F(3,35) = 6.02, *p* = 0.002), with significantly more time in the APP/PS1 (*p* < 0.01) and APP/PS1/eNOS^+/−^ (*p* < 0.05) groups when compared to the WT group (Figure 2B). Regarding the duration of rearings, we found no significant genotype effect (F = 1.8; data not shown). There were no significant differences between APP/PS1 and APP/PS1/eNOS^+/−^ mice in all these measurements. Regarding the elevated plus maze, we found no significant effect of genotype in the time spent in open or closed arms (both F < 1; Figure 2C,D, respectively). Animals were then tested in the Y-maze apparatus on day two. There were no significant effects of genotype in the total number of arm entries (F(3,35) = 2.66, *p* = 0.067; Figure 2E) and % spontaneous alternation (F(3,35) = 2.36, *p* = 0.09; Figure 2F). These data demonstrated that mice with APP and PS1 mutations were hyperactive, and partial eNOS deficiency did not affect locomotion, exploration, anxiety, and working memory in APP/PS1 mice. 

On day three, the animals in all four genotype groups had a 60 s swimming session in the water maze with no platform presented and had no signs of difficulty in swimming. All animals were then tested in a working memory version of the water maze task from day four for 3 consecutive days. On each day, animals were trained to find a visible platform (trial one) and a hidden platform placed in the same position (trials two–six). The data obtained during trial one (visible platform) or trials two–six (hidden platform) over the three days of testing were averaged separately. Regarding the swimming speed (Figure 3A), there were no significant genotype effects in both testing conditions (all F < 2). In terms of the path length measurement (Figure 3B), ANOVA revealed a significant genotype effect under the condition of hidden platform (F(3,36) = 12.04, *p* < 0.0001), but not visible platform (F(3,36) = 2.19, *p* = 0.11). Post hoc tests confirmed a significant longer path length to reach the hidden platform in APP/PS1 mice relative to WT controls (*p* < 0.05), and in APP/PS1/eNOS^+/−^ mice when compared to WT mice (*p* < 0.0001), eNOS^+/−^ mice (*p* < 0.001) and APP/PS1 mice (*p* < 0.05). When the percentage of path length in the outer zone (thigmotaxis) was analyzed (Figure 3C), there were significant genotype effects under the conditions of visible platform (F(3,36) = 5.30, *p* = 0.0039) and hidden platform (F(3,36) = 13.22, *p* < 0.0001). Under the visible platform condition, although both APP/PS1 and APP/PS1/eNOS^+/−^ mice generated greater thigmotaxis, post hoc tests confirmed statistical significances between the APP/PS1 group and the WT (*p* < 0.01) and eNOS (*p* < 0.05) groups. Under the hidden platform condition, APP/PS1 and APP/PS1/eNOS^+/−^ mice generated greater thigmotaxis when compared to WT mice (both *p* < 0.001) and eNOS^+/−^ mice (both *p* < 0.01). Since there was no significant difference between the APP/PS1 and APP/PS1/eNOS^+/−^ groups in thigmotaxis (Figure 3C), the longer path length in APP/PS1/eNOS^+/−^ mice suggest that partial eNOS deficiency exacerbated spatial learning deficits in APP/PS1 mice. 

### 2.3. Amyloid Deposition in the Brain

Since APP/PS1/eNOS^+/−^ mice displayed poorer spatial learning relative to APP/PS1 mice, immunofluorescence with 6E10 antibody was employed to determine how partial eNOS deficiency affected Aβ deposition in the brain in APP/PS1 mice at 8 months of age. Immunofluorescence revealed no 6E10 immunoreactivity in the hippocampus or cortex in WT and eNOS^+/−^ mice, as well as APP/PS1 and APP/PS1/eNOS^+/−^ mice under the condition of primary antibody omission (data not shown). By contrast, however, the widespread 6E10 positive staining was clearly present in the coronal sections of the hippocampus and cortex at the levels of AP −1.70, −2.30, and −2.92 mm in both APP/PS1 (Figure 4A,D,G) and APP/PS1/eNOS^+/−^ (Figure 4B,E,H) mice. Morphologically, a combination of small and compact and/or diffuse and dense-core amyloid plaques could be seen in both APP/PS1 and APP/PS1/eNOS^+/−^ groups, although the density varied between regions. When comparing the 6E10 immunoreactive profiles between the two genotype groups, there was a clear pattern of more Aβ plaques in both the cortex and hippocampus at each level in APP/PS1/eNOS^+/−^ mice relative to APP/PS1 mice. We then quantified the Aβ load in the cortex and hippocampus by measuring the percentage area occupied by Aβ plaques at each level for each animal. APP/PS1/eNOS^+/−^ mice displayed significantly higher Aβ loads in both the cortex (44–98% increases) and hippocampus (140–170% increases) relative to APP/PS1 mice at the levels of AP −1.70 (both *p* = 0.0095; Figure 4C), −2.30 (both *p* = 0.038; Figure 4F) and −2.92 (CX: *p* = 0.016; HP: *p* = 0.0043; Figure 4I). These findings demonstrated that partial eNOS deficiency increased Aβ load in the brains of APP/PS1 mice. 

### 2.4. Microglial Immunoreactivity and Colocalization with Aβ

Because of the well-established role of microglia in limiting Aβ deposits through phagocytosis and degradation [21,22], immunofluorescence with ionized calcium-binding adaptor molecule 1 (Iba-1, a macrophage/microglia-specific protein) antibody was then employed to determine the immunoreactivity of microglia in the hippocampus of WT, eNOS^+/−^, APP/PS1 and APP/PS1/eNOS^+/−^ mice. Moreover, double immunofluorescent labeling was used to determine the colocalization of microglia and Aβ using Iba-1 and 6E10 antibodies. Immunofluorescence with the Iba-1 antibody revealed a widespread distribution of microglia in the hippocampus in both WT (Figure 5B) and eNOS^+/−^ mice (Figure 5F). At high magnification, microglia appeared to have small soma and extensively ramified processes in WT (Figure 5D-a) and eNOS^+/−^ mice (Figure 5H-a), consistent with the morphology of resting microglia. Again, there was no 6E10 immunoreactivity in the hippocampus of WT (Figure 5C) and eNOS^+/−^ mice (Figure 5G). By comparison, however, intensely stained microglia forming clusters were evident in the hippocampus of APP/PS1 (Figure 5J) and APP/PS1/eNOS^+/−^ mice (Figure 5N). In addition to resting microglia with ramified morphology, clustered microglia with enlarged soma and shortened processes were evident in APP/PS1 (Figure 5L-a and Figure 6A-a1) and APP/PS1/eNOS^+/−^ mice (Figure 5P-a and Figure 6B-a1), indicative of the activated microglia phenotype. Immunofluorescence with the 6E10 antibody revealed the presence of Aβ plaques in the hippocampus in APP/PS1 (Figure 5K) and APP/PS1/eNOS^+/−^ mice (Figure 5O), again with more plaques in the latter genotype group. The merged images of 6E10 and Iba-1 double immunofluorescent labeling clearly showed the colocalization of amyloid plaques and microglial clusters in both APP/PS1 (Figure 5L) and APP/PS1/eNOS^+/−^ mice (Figure 5P). Under high magnification, microglia with large soma and short processes colocalized with amyloid plaques (surrounding the core of the plaques) and microglial processes appeared to interact directly with the plaques in both APP/PS1 (Figure 5L-a) and APP/PS1/eNOS^+/−^ mice (Figure 5P-a).

As described above, immunofluorescence with the Iba-1 antibody revealed microglia surrounding plaques with an enlarged soma and shortened processes in the hippocampus in both APP/PS1 and APP/PS1/eNOS^+/−^ mice, indicative of the activated phenotype. The cluster of differentiation 68 (CD68) is a transmembrane glycoprotein that labels the lysosome and therefore indicates the phagocytic activity of activated microglia. We, therefore, carried out double immunofluorescent labeling using CD68 and Iba-1 antibodies to determine the status of microglial activation in the hippocampus. In both the WT and eNOS^−/+^ groups, there was no obvious CD68 positive staining (data not shown). By comparison, however, scattered CD68 positive staining forming clusters was evident in both APP/PS1 (Figure 6A) and APP/PS1/eNOS^+/−^ mice (Figure 6B). Moreover, CD68 and Iba-1were almost always co-expressed in microglia with amoeboid morphology in both genotypes (Figure 6A-a,B-a). The immunoreactivity of CD68 confirmed the phagocytic activity of activated microglia in the hippocampus in both APP/PS1 and APP/PS1/eNOS^+/−^ mice.

### 2.5. Concentration of Soluble Aβ in the Brain

The results of 6E10 immunofluorescent labeling indicate partial eNOS deficiency increased amyloid plaque load (insoluble Aβ) in the brains of APP/PS1 mice. Using enzyme-linked immunosorbent assay (ELISA) kits, we measured the concentrations of the soluble fractions of Aβ_1–40_ and Aβ_1–42_ in the anterior cortex of APP/PS1 and APP/PS1/eNOS^+/−^ mice (data expressed as ng/mg protein). Mann–Whitney U test revealed no significant differences between the two genotype groups in soluble Aβ_1–40_ (APP/PS1: 9.83 ± 1.16; APP/PS1/eNOS^+/−^: 10.72 ± 0.69; *p* = 0.24) and Aβ_1–42_ (APP/PS1: 1.28 ± 0.13; APP/PS1/eNOS^+/−^: 1.46 ± 0.07; *p* = 0.44).

### 2.6. Protein Expression

Immunofluorescence with 6E10 antibody revealed significantly increased Aβ plaque load in 8 months old APP/PS1/eNOS^+/−^ mice relative to their age-matched APP/PS1 mice (Figure 4). In order to investigate the molecular mechanisms underlying increased Aβ plaque deposition in the brain due to partial eNOS deficiency, we determined protein expression of enzymes involved in Aβ synthesis (BACE-1) and clearance (such as LRP-1, IDE, AQP4) as well as markers of microglia (Iba-1) and astrocytes (GFAP) in the anterior cortex of APP/PS1 and APP/PS1/eNOS^+/−^ mice via western blot. Mann–Whitney U test revealed significantly increased levels of BACE-1 (36% increase; *p* = 0.0091; Figure 7A,G), LRP-1 (31% increase; *p* = 0.049; Figure 7B,G) and Iba-1 (31% increase; *p* = 0.021; Figure 7E,G), but significantly reduced level of IDE (35% decrease; *p* = 0.0002; Figure 7C,G), in APP/PS1/eNOS^+/−^ mice relative to APP/PS1 mice. Although there was a clear trend of an increased level of AQP4 in APP/PS1/eNOS^+/−^ mice (22% increase; Figure 7D,G), this difference was not statistically significant (*p* = 0.087). Similarly, a 22% increase in GFAP protein was found in APP/PS1/eNOS^+/−^ mice relative to APP/PS1 mice (Figure 7F,G), which, however, did not reach the statistical significance (*p* = 0.14). When taken together, these data demonstrated that partial eNOS deficiency altered the enzymes involved in Aβ formation and clearance, leading to increased Aβ burden and microglial activation in the brain of APP/PS1 mice. 

## 3. Discussion

Currently, there are no effective therapies for AD, the most common dementia in the aged, despite decades of trials targeting Aβ. Therefore, there is an urgent need to better understand other mechanisms underlying the disease process to identify novel therapeutic targets and to create new animal models that better recapitulate the human disease state. Recent research has implicated cerebral endothelial dysfunction in the pathogenesis of AD [11,12,13,14,15]. eNOS deficiency is evident in the AD brains and is inversely correlated with amyloid plaques and NFTs [9,10,11]. Moreover, eNOS deficiency alters APP processing and promotes AD pathology [12,13,14,15,16]. The present study generated the APP/PS1/eNOS^+/−^ mouse model of AD by adding partial eNOS deficiency to APP/PS1 mice. APP/PS1/eNOS^+/−^ mice at 8 months of age did not show significant differences in body weight or organ/body weight ratios when compared to WT, eNOS^+/−^, and APP/PS1 mice. Moreover, partial eNOS deficiency did not affect the survival of APP/PS1 mice up to 8 months of age. We further characterized APP/PS1/eNOS^+/−^ mice with a focus on cognitive function, Aβ deposition in the brain and microglial pathology, and the enzymes involved in Aβ production and clearance. 

Animals in all four genotype groups were assessed in a battery of behavioral tasks. The open field assesses animals’ locomotion and exploratory activity, as rodents display natural conflict between exploration of and aversion to bright open areas in a novel environment [23]. APP/PS1 mice were hyperactive, as they generated a significantly longer path length relative to WT and eNOS^+/−^ mice (although not APP/PS1/eNOS^+/−^ mice). Both APP/PS1 and APP/PS1/eNOS^+/−^ groups spent significantly more time in the outer zone of the apparatus relative to the WT group, indicative of a higher level of anxiety in mice with APP and PS1 mutations. There was no significant genotype difference in the exploratory activity based on the rearing measurement. The elevated plus maze is a commonly used test to assess the anxiety level in rodents based on their natural aversion to open and elevated areas [23,24,25]. Interestingly, we found no significant differences between groups in the time spent in open or closed arms, suggesting no genotype-related changes in anxiety. Spontaneous alternation behavior in the Y-maze is an index of spatial working memory in rodents, which is based on their innate curiosity to explore previously unvisited arms [26,27]. The present study found no significant differences between groups in terms of the percentage of spontaneous alternation and the total number of arm entries, indicating that APP and PS1 mutations did not affect working memory and locomotion in this test. Earlier research has reported mixed results in these tasks in APP/PS1 mice [28,29,30,31,32,33,34,35]. However, it is currently unclear what contributes to the discrepancies. 

The water maze is the most commonly used test to assess spatial learning and memory in rodents, and the integrity of the hippocampus is essential for this task [36]. We have previously reported that APP/PS1 mice were significantly impaired in the working memory version of the water maze task [20]. Using the same experimental protocol, we found that APP/PS1 mice generated significantly longer path length to reach the hidden platform, but not the visible platform, relative to WT mice. Intriguingly, the APP/PS1/eNOS^+/−^ group was severely impaired in this task when compared to the WT and eNOS^+/−^ groups and performed significantly worse than the APP/PS1 group based on the path length measurement to the hidden platform. Since there were no significant differences between the APP/PS1 and APP/PS1/eNOS^+/−^ groups in the swimming speed and thigmotaxis measurements, the longer path length generated by APP/PS1/eNOS^+/−^ mice indicates poorer spatial working memory in relation to APP/PS1 mice. When taking all behavioral data together, the present study demonstrated that partial eNOS deficiency aggravated the spatial working memory deficit in APP/PS mice, but did not affect locomotion, exploration, anxiety, and spontaneous alternation behavior. 

APP/PS1 mice display progressive Aβ deposition in the brain, with clear amyloid plaque pathology at 8 months of age [37]. The hippocampus plays a critical role in learning and memory processing [38,39] and is also one of the earliest and the most intensely affected brain regions in AD [40,41,42]. Given the role of eNOS-derived NO in modulating APP processing and in maintaining cerebrovascular function [12,13,16], we postulated that partial eNOS deficiency might exacerbate amyloid plaque load and other AD-like pathology in the hippocampus of APP/PS1 mice. Immunofluorescence with 6E10 antibody revealed the presence of small compact and diffuse dense-core amyloid plaques in the hippocampus and cortex in both the APP/PS1 and APP/PS1/eNOS^+/−^ groups, with a clear pattern of more Aβ plaques in APP/PS1/eNOS^+/−^ mice. When the percentage area covered by Aβ plaques was analyzed, APP/PS1/eNOS^+/−^ mice displayed significantly high Aβ loads at all three levels of the cortex (44–98% increases) and hippocampus (140–170% increases) when compared to APP/PS1 mice. However, ELISA revealed no significant differences between the two genotype groups in the soluble fractions of Aβ_1–40_ and Aβ_1–42_. 

Aβ is produced by the amyloidogenic processing of APP, which involves both β-secretase (BACE1) and γ-secretase [43]. It has been shown that the BACE1 promoter and the 5′ untranslated region contain the binding sites for cAMP response element binding protein (CREB), nuclear factor κB (NFκB), stimulating protein (Sp) 1 site, and yin yang (YY)1, which can be modified by NO [44,45,46,47,48]. Earlier research has reported that eNOS^−/−^ mice at both young and older ages have increased levels of APP, BACE1, Aβ_1–40_, and Aβ_1–42_ in the brain and cerebral microvessels relative to their age-matched WT controls [12,13,17]. Nitroglycerin can be converted to NO in the brain, and its supplementation attenuates the upregulation of APP and BACE1 protein levels and the increases in Aβ in cerebral microvessels of eNOS^−/−^ mice [17]. In concert with this, other studies have documented that NO can suppress BACE1 levels [49,50]. The present study found a significantly increased level of BACE1 in APP/PS1/eNOS^+/−^ mice relative to APP/PS1 mice, further supporting the role of endothelial NO in modulating the amyloidogenic processing of APP.

Aβ in the brain can be cleared via various mechanisms, such as enzymatic degradation, interstitial fluid bulk flow, and transportation to the blood. IDE, a metalloprotease highly expressed in the brain, involves the degradation of Aβ polymers and fibers and is a major regulator of Aβ levels in neuronal and microglial cells [51,52,53,54]. In AD brains, the levels of IDE expression diminish in the hippocampus and cortex as a function of age [55]. Interestingly, the size of amyloid plaques correlates negatively with IDE expression and activity [56], indicating the involvement of lDE deficiency in plaque build-up. The present study found significantly reduced levels of IDE protein in APP/PS1/eNOS^+/−^ mice relative to APP/PS1 mice, which is likely attributed to higher Aβ load in the former. Aβ can also be cleared from the brain by the glymphatic system through AQP4 water channels in astrocytes [57,58]. We observed a 22% increase in both AQP4 and GFAP protein expression in APP/PS1/eNOS^+/−^ mice relative to APP/PS1 mice, although the differences were not statistically significant. LRP-1 is highly expressed in a variety of cell types in the brain and plays a critical role in maintaining brain homeostasis and controlling Aβ metabolism [59,60,61,62,63]. LRP-1 can influence both the clearance and production of Aβ, and vascular LRP-1 plays an important role in transporting Aβ from the brain to blood [59,60,61,62,63]. LRP-1 is also highly expressed in reactive astrocytes and activated microglia associated with mature amyloid plaques [64,65,66]. The present study found significantly increased LRP-1 protein levels in APP/PS1/eNOS^+/−^ mice relative to APP/PS1 mice, which may be a compensatory adaptation to combat rising Aβ levels [14]. Since our data only represented the net LRP-1 protein increase in the lysate of the cortex, it is of interest to investigate how LRP-1 expression changes in different compartments of the brain in the future. When taken together, partial eNOS deficiency exacerbated Aβ pathology in APP/PS1 mice by augmenting the amyloidogenic processing of APP and reducing Aβ clearance. 

Microglia are the innate immune cells in the brain and act as the first and main defense against noxious stimuli. In AD brains, activated microglia migrate to the plaque areas aiming to phagocytose and eliminate the plaques [67,68,69]. Immunofluorescence with Iba-1 (microglial marker) revealed a widespread distribution of microglia with small soma and extensively ramified processes (indicating the morphology of resting microglia) in the hippocampus of WT and eNOS^+/−^ mice. In the APP/PS1 and APP/PS1/eNOS^+/−^ mice, however, there were also intensely stained microglia forming clusters with enlarged soma and shortened processes. Immunofluorescent double labeling with Iba-1 and 6E10 antibodies showed that these clustered intensely stained microglia were colocalized with amyloid plaques and appeared to surround the core of the plaques and/or interact directly with the plaques in both the APP/PS1 and APP/PS1/eNOS^+/−^ mice. Earlier research has reported a strong correlation between plaque burden and microglial gene expression, encompassing both proliferation and activation of microglia [70], and NO can modulate microglial activation and migration [71,72]. It should be pointed out that the present study did not quantify the load of clustered microglia. Given the colocalization of microglia and amyloid plaques and the Aβ load quantification results described above, however, partial eNOS deficiency appeared to augment microglial pathology in APP/PS1 mice. This conclusion was further supported by the western blot data showing a significantly increased Iba-1 protein level in APP/PS1/eNOS^+/−^ mice relative to APP/PS1 mice. Iba-1 has been implicated in cell skeleton reorganization, phagocytosis, and motility of microglia [73,74]. The clustered microglia with enlarged soma and shortened processes indicate that microglia associated with plaques were at the reactive status in both APP/PS1 and APP/PS1/eNOS^+/−^ mice. CD68, a specific marker for activated microglia [75], is primarily expressed in amoeboid microglia with little expression in resting microglia [76]. Immunofluorescent double labeling using Iba-1 and CD68 revealed almost negligible expression of CD68 in WT and eNOS^+/−^ mice. However, the clustered CD68 and Iba1 immunoreactive microglia were evident in both the APP/PS1 and APP/PS1/eNOS^+/−^ mice, indicative of a phagocytic phenotype of microglia. It has been shown that Iba-1 is often upregulated during microglial activation, whereas CD68 is upregulated during phagocytosis [76]. Moreover, the consistent CD68 upregulation is positively correlated with AD pathology and cognitive impairment [77]. 

In summary, the present study, for the first time, demonstrated that APP/PS1/eNOS^+/−^ mice (the APP/PS1 mice with partial eNOS deficiency) at 8 months of age display impaired spatial learning and increased Aβ plaque load and Iba-1 expression in the brain relative to APP/PS1 mice. As illustrated in Figure 8, partial eNOS deficiency augmented the amyloidogenic processing of APP (upregulated BACE-1), reduced cellular Aβ clearance (downregulated IDE) and system Aβ clearance (upregulated LRP-1, speculatively a compensatory response [14]), and exacerbated microglial pathology (likely increased inflammation) in APP/PS1 mice. These changes, along with other mechanisms, ultimately augmented cognitive dysfunction in APP/PS1 mice. Given the marked phenotype differences in APP/PS1 mice with and without partial eNOS deficiency at 8 months of age, future research is required to determine whether partial eNOS deficiency alters the onset of cognitive deficits or Aβ deposition in the brain in APP/PS1 mice. Since female APP/PS1 mice display more severe AD pathology relative to males [78], it is essential to characterize the sex differences in APP/PS1/eNOS^+/−^ mice in the future. In conclusion, our findings further support the role of endothelial dysfunction in the pathogenesis of AD. APP/PS1/eNOS^+/−^ mice offer a more clinically relevant model for early-stage AD with an element of endothelial dysfunction to better understand the disease pathogenesis and to develop preventive and/or therapeutic strategies. 

## 4. Materials and Methods

### 4.1. Animals

APP/PS1 (B6.Cg-Tg(APPswe,gPSEN1dE9)85Dbo/Mmjax; https://www.jax.org/strain/005864; accessed on 15 December 2014) and wildtype mice on C57BL/6J background, and eNOS^−/−^ mice (Nos3tm1Unc/J; https://www.jax.org/strain/002684; accessed on 15 December 2014) were originally obtained from Jackson laboratories. Male APP/PS1 mice and female wildtype littermates were crossed to produce wildtype (WT) and APP/PS1 offspring (Figure 1A), whereas male eNOS^−/−^ and female APP/PS1 mice were crossed to produce eNOS^+/−^ and APP/PS1/eNOS^+/−^ mice (Figure 1B). Ear notching samples were collected from offspring on postnatal days (PND) 14–21. The genotype of offspring was based on the absence or presence of APPswe and PSEN1dE9 mutations. Partial eNOS deficiency in eNOS^+/−^ and APP/PS1/eNOS^+/−^ mice was confirmed by western blotting (Figure 1B). 

Animals at 8 months of age were used in the present study. In Experiment 1, male WT, eNOS^+/−^, APP/PS1, and APP/PS1/eNOS^+/−^ mice (*n* = 8–12/group) were assessed behaviorally, followed by tissue collection for western blotting to evaluate genotype-related changes in protein expression. In Experiment 2, a separate set of male WT, eNOS^+/−^, APP/PS1, and APP/PS1/eNOS^+/−^ mice from the same cohort (*n* = 5–6/group) were used for immunohistochemistry to determine the effects of partial eNOS deficiency on amyloid deposition and microglia in the brain of APP/PS1 mice. All animals were housed individually (13 × 15 × 38 cm^3^), maintained on a 12 h light/dark cycle (lights on at 7 AM), and provided ad libitum access to food and water. Animals’ body weights and general health conditions were closely monitored. All experimental procedures were carried out in accordance with the regulations of the University of Otago Animal Ethics Committee and the Animal Research: Reporting of In Vivo Experiments (ARRIVE) guidelines. Every attempt was made to reduce the number of animals used and to minimize their suffering.

### 4.2. Behavioural Procedures

In Experiment 1, all animals were behaviorally tested in the open field, elevated plus maze, Y-maze, and water maze. All behavioral experiments were conducted in a windowless room. A video camera was placed in the center of the room at ceiling height to record animals’ behavioral performance, and a radio speaker was located adjacent to the video camera to provide background masking noise. The extramaze cues (the laboratory furniture, lights, and several prominent visual features on the walls, as well as the location of the experimenter) were held constant throughout the entire experiment. The experimenter was blind to the grouping information at the time of behavioral testing and analysis. 

#### 4.2.1. Open Field (Day 1)

The open field apparatus was a 40 × 40 cm white Plexiglas box with identical 20 cm high walls and an open top and was elevated approximately 60 cm from the floor. On day 1, each animal was placed into the chamber at the same position and allowed to explore the apparatus freely for 5 min. The order of animal testing was counterbalanced between the four genotype groups. Animal behavior was recorded and analyzed offline using TopScan (Cleversys Inc. Reston, VA, USA). The total path length traveled, the duration of rearing, and the percentage of time spent in the outer zone (10 cm from the wall) were analyzed [28,79].

#### 4.2.2. Elevated plus Maze (Day 1)

The elevated plus maze had four arms (29 × 6 cm), two open arms surrounded by 1 cm clear Plexiglas and two closed arms surrounded by 15 cm high white Plexiglas, and a central area (6 × 6 cm). Again, the apparatus was elevated 60 cm above the floor. After completion of the open field test, each mouse was placed in the center of the maze facing one of the closed arms and allowed to explore the apparatus freely for 5 min. The order of animal testing was counterbalanced between the four genotype groups. The total number of arm entries and the time spent in the open or closed arms were recorded. An arm entry was scored when all four paws were in the arm [28,79].

#### 4.2.3. Y-Maze (Day 2)

The Y-maze was shaped like a Y and made of white Plexiglas with a 120° angle between each of the three arms (40 × 6 × 13 cm). The apparatus was elevated 60 cm above the floor, and the positions of the arms were kept constant for all animals during the test. Each arm was assigned either A, B, or C. On day 2, mice in four genotype groups were individually placed at the center of the maze facing the arm ‘A’ and allowed to freely explore the apparatus for a period of 5 min. The order of animal testing was counterbalanced between groups. The number and the sequence of arm entries were recorded. Again, an arm entry was scored when all four paws were in the arm. Alternation behavior was defined as consecutive entries into all three arms (i.e., ABC, CAB, or BCA, but not ABA). The percentage of spontaneous alternations was calculated as the ratio of the actual number of alternations to the possible number (defined as the total number of arm entries minus two) multiplied by 100, i.e., % alternation = [(number of alternations)/(total number of arm entries − 2)] × 100 [80,81]. The percentage of spontaneous alternation was measured as an index of working memory, whereas the total number of arm entries reflected the level of locomotor activity [79,80].

#### 4.2.4. Water Maze (Days 3–6)

The water maze was a white plastic circular tank measuring 100 cm in diameter and 35 cm in height. It was filled daily to the level of approximately 15 cm below the top, and the water temperature was maintained at 22  ±  1 °C. Four points on the edges of the tank were designated as north (N), south (S), east (E), and west (W) so that the pool was divided into the NE, SW, NW, and SE quadrants. Animals’ swimming ability was tested on day 3. Each mouse was placed into the pool facing the wall from the starting point W and was allowed to swim freely for a duration of 60 s. 

From day 4, all animals were tested in a working memory version of the water maze task for 3 consecutive days [20,79], and the order was counterbalanced across the four genotype groups. On each day, there were 6 trials for each mouse with a 90 s interval between trials. For the first trial, the animal was trained to find a visible platform (6 cm in diameter), which was 1 cm above the water with the edge attached to a black upright plastic tag (5 × 3 cm^2^). For the subsequent 5 trials (trials 2–6), the animal was trained to find a hidden platform that was placed in the same position, however, 2 cm below the water. For each trial, the mouse was placed into the pool facing the wall and allowed to swim freely in search of the platform for a maximum of 60 s and to stay on the platform for 10 s before being removed, dried, and kept warm in a holding box. If the mouse did not find the platform within 60 s of being placed into the pool, it was immediately placed on or near-guided to the platform for 10 s before being returned to the holding box. The platform location and starting points (N, S, W, E) were pseudo-randomly selected and changed on each day but were kept the same for all of the animals. 

Animals’ behavior was recorded, and several performance variables were analyzed offline using TopScan. The distance (path length) the mouse swam during the 60 s period of the swim test (day 3) or from the starting point to reach the visible/hidden platform traveled prior to finding the platform (days 4–6), the degree of thigmotaxic swimming (quantified by dividing the maze into two circles and measuring the time spent in the outer 10% of the pool) and the swimming speed were measured [20,79]. 

### 4.3. Organ Collection and Brain Tissue Preparation

For Experiment 1, the day after completion of the water maze test, all animals were transcardially perfused with ice-cold saline. The brains were rapidly removed and left in cold saline (4 °C) for at least 45 s and then weighed. Following a mid-sagittal cut, the anterior portion of the cerebral cortex was freshly dissected at the level of anterior commissure from each hemisphere on ice, snap-frozen, and then stored at −80 °C for western blotting. The major organs (such as liver, kidney, spleen, and heart) were collected from each animal and weighed. 

For Experiment 2, all animals were transcardially perfused with ice-cold saline followed by 4% phosphate-buffered paraformaldehyde. Whole brains were removed, fixed in 4% paraformaldehyde for a further 24 h, and then left in 30% sucrose at 4 °C until they sunk. The forebrain was divided into two hemispheres, which were embedded separately in optimal cutting temperature compound. Each hemisphere was then rapidly frozen using liquid nitrogen cooled isopentane and stored at −80 °C until sectioning for immunohistochemistry.

### 4.4. Immunofluorescence Procedures

Serial coronal sections (30 µm) were cut for each hemibrain using a cryostat (Leica CM 1950) and stored at −20 °C in cryoprotectant (30% vol/vol glycerol, 30% vol/vol ethyl glycol, 20% vol/vol 0.1 M phosphate buffer and 20% vol/vol H_2_O). Brain sections at the levels of AP −1.70, −2.30, and −2.92 mm (based on the mouse brain atlas [81]) were selected from each animal for immunofluorescence to compare the amyloid plaque load (6E10) and microglial immunoreactivity (Iba-1) in the brain between groups. The sections from all animals were run at the same time and under the same experimental conditions. Unless stated, the incubations occurred at room temperature and were followed by triplicate washes in 1% phosphate-buffered saline (PBS) on an orbital shaker.

Briefly, the sections were incubated in PBS with 0.1 M glycine for 10 min to remove any aldehyde radicals from fixation. In order to expose the epitopes of the Aβ peptides within the plaques for the 6E10 antibody detection, the sections were incubated with 90% formic acid for 7 min and then blocked with 10% normal goat serum in incubation buffer (0.3% Triton in 1% bovine serum albumin/PBS) for 1 h. The sections were incubated at 4 °C overnight with a cocktail of primary antibodies containing either monoclonal mouse antibody raised against 6E10 (1:1000, Covance, sig-39320-50), polyclonal rabbit antibody raised against Iba-1 (1:1000, Wako, 019-19741) or monoclonal mouse antibody raised against CD68 (1:500, Abcam, ab31630), all diluted in incubation buffer. On the following day, after extensive rinsing, sections were incubated with a cocktail of florescent secondary antibodies: Alexa 488 goat ant-rabbit (A11070) and Alexa 594 goat anti-mouse (both 1:1000, Molecular probes, A11020). Hoescht (1:1000) was then applied for 10 min to stain the cell nuclei. The sections were mounted onto gelatin slides and coverslipped using 1,4-diazabicyclo [2.2.2]octane (DABCO) glycergel. Immunohistochemical controls were taken by omitting the primary antibodies.

### 4.5. Imaging and Amyloid Plaque Quantification

Sections were visualized using Nikon Ti2E inverted fluorescence montaging microscope attached to a camera (Nikon DSiR2 color 16MP), and NIS-Elements digital microscopy software was used to capture images. Images were captured with constant exposure time, offset and gain for each staining marker. Two brain regions of interest, the entire cortex (CX) and hippocampus (HP), were manually delineated. The percentage area of positive signal for 6E10 was measured using 8-bit images. The threshold value was maintained for all images, and the percentage area covered by 6E10 positive plaques was calculated using the ImageJ algorithm [82].

### 4.6. Enzyme-Linked Immunosorbent Assay

The concentrations of the soluble fractions of Aβ_1–40_ and Aβ_1–42_ in the anterior cortex of APP/PS1 and APP/PS1/eNOS^+/−^ mice were measured using ELISA kits: the human amyloid β (1–40) assay kit (IBL, Japan, 27,713) and the human amyloid β (1–42) assay kit (IBL, 27,711) according to manufacturer’s instructions. Each assay was performed in duplicate, and the sample order was counterbalanced between the APP/PS1 and APP/PS1/eNOS^+/−^ groups. The levels of soluble Aβ_1–40_ and Aβ_1–42_ were expressed as ng/mg protein [82].

### 4.7. Western Blot

The protein expression of eNOS, BACE1, LRP1, IDE, GFAP, AQP4 or Iba-1 in each sample was determined using western blots. The cortical tissue samples were lysed in 50 mM Tris-HCl (pH 7.4) supplemented with a cocktail of protease (15 µm pepstatin A, 2 µm leupeptin, and 10 µm phenylmethylsulfonyl fluoride,) and phosphatase inhibitors by sonication on ice. The protein concentration in each sample was determined by Bradford assay and then equalized to 2 mg/mL.

Brain tissue homogenates were mixed with gel loading buffer containing 50 mM Tris-HCl, premixed XT sample buffer (Bio-Rad Laboratories, Ref. 1610791), and reducing agent (Bio-Rad Laboratories, Ref. 11610792) in a ratio of 1:1, and then boiled for 5 min. The samples (4–15 μL), along with pre-stained protein marker and a biological control sample, were loaded in each well on a Criterion^TM^ XT 4–12% gradient (Bio-Rad; 3450125) or 12% SDS-PAGE gel (Bio-Rad; 3450119) in a counterbalanced manner and electroblotted onto a nitrocellulose membrane (Bio-Rad) as detailed in our previous publications [80,81]. The membranes were blocked with 5% BSA in Tris-buffered saline (pH 7.4) containing 1% Tween 20 (TBST) for 4 h, and then incubated with primary mouse monoclonal antibody against eNOS (1:500, BD Biosciences, 610297) or GFAP (1:100,000, Sigma, G3893), rabbit monoclonal antibody against BACE1 (1:1000, Cell signalling, D10E5), LRP1 (1:5000, Abcam, ab 92544) or GAPDH (1:100,000, Abcam, ab 181602), or rabbit polyclonal antibody against IDE (1:2000, Abcam, ab32216), AQP4 (1:2000, Millipore, AB3594) or Iba-1 (1:500; Wako, 019-19741) overnight at 4 °C. The membranes were then probed with IRDye^®^ 680RD goat anti-mouse IgG antibody or IRDye^®^ 800 CW donkey anti-rabbit IgG antibody (both 1:10,000, LI-COR Biosciences, Lincoln, NE, USA) for 4 h. Detection of the immunoreactive signal bands was performed using Odyssey^®^ CLx Imager (LI-COR, Lincoln, NE, USA). Signals were quantified with Odyssey CLx Image Studio software (LI-COR Biosciences) and normalized by the corresponding GAPDH loading controls and biological controls to account for inter-gel variation. The experimenters were blind to the grouping information at the time of the assay and analysis.

### 4.8. Statistical Analysis

All the behavioral results and eNOS protein expression data obtained from four genotype groups were analyzed using one-way analysis of variance (ANOVA) followed by post hoc tests. Amyloid plaque load, soluble Aβ_1–40_ and Aβ_1–42_, and western blotting data obtained from the APP/PS1 and AP/PS1/eNOS^+/−^ groups were analyzed using Mann–Whitney U test. Statistical analyses were performed using GraphPad Prism software (Version 9.3.1). All data were presented as mean and standard error of the mean (mean ± SEM), and the level of significance was set at *p* < 0.05 for all comparisons.

## Figures and Tables

**Figure 1 ijms-23-07316-f001:**
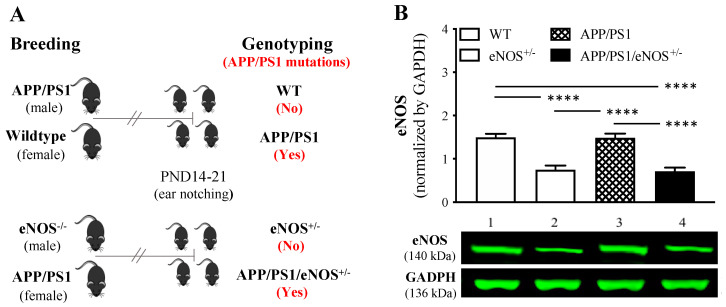
(**A**) Diagrams illustrating the breeding and genotyping scheme. Male APP/PS1 mice and female wildtype littermates were crossed to produce wildtype (WT) and APP/PS1 offspring, and the genotype of offspring was based on the absence (WT mice) or presence (APP/PS1 mice) of APPswe (APP) and PSEN1dE9 (PS1) mutations. Male eNOS^−/−^ and female APP/PS1 mice were crossed to produce eNOS^+/−^ and APP/PS1/eNOS^+/−^ offspring, and the genotype of offspring was based on the absence (eNOS^+/−^ mice) or presence (APP/PS1/eNOS^+/−^ mice) of APP and PS1 mutations. Ear notching samples were collected from offspring on postnatal days (PND) 14–21 for genotyping. (**B**) mean (± SEM) eNOS protein levels (arbitrary unit, normalized by glyceraldehyde 3-phosphate dehydrogenase, GAPDH) in the anterior cortex of WT, eNOS^+/−^, APP/PS1 and APP/PS1/eNOS^+/−^ mice (*n* = 8–12/genotype). Representative western blots of eNOS and GAPDH in WT (1), eNOS^+/−^ (2), APP/PS1 (3) and APP/PS1/eNOS^+/−^ mice (4). **** indicates a significant effect between groups at *p* < 0.0001.

**Figure 2 ijms-23-07316-f002:**
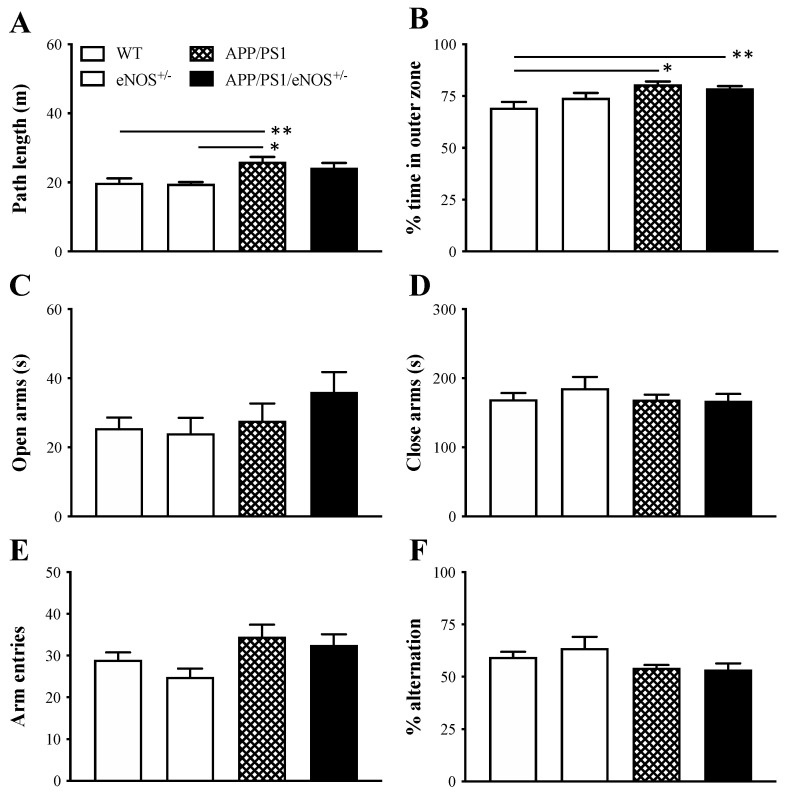
Mean (± SEM) path length (**A**) and % time in the outer zone (**B**) in the open field, time (s) spent in the open arms (**C**) and close arms (**D**) in the elevated plus maze, and the total number of arm entries (**E**) and the percentage of spontaneous alternation (**F**) in the Y-maze in 8-months old wildtype (WT), eNOS^+/−^, APP/PS1 and APP/PS1/eNOS^+/−^ mice (*n* = 8–12/genotype). * indicates a significant difference between groups at * *p* < 0.05 or ** *p* < 0.01.

**Figure 3 ijms-23-07316-f003:**
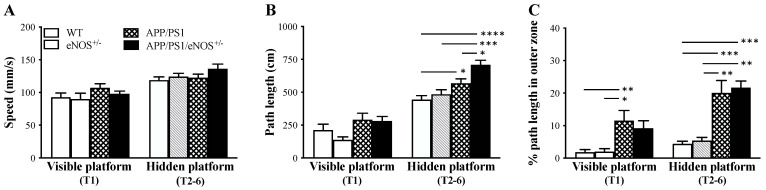
Mean (± SEM) speed (**A**), path length to reach the platform (**B**) and percentage of path length in the outer zone (thigmotaxic swimming; (**C**) during trial one (T1; visible platform) and trials two–six (T2–6; hidden platform) in 8-months old wildtype (WT), eNOS^+/−^, APP/PS1 and APP/PS1/eNOS^+/−^ mice (*n* = 8–12/genotype). * indicates a significant difference between groups at * *p* < 0.05, ** *p* < 0.01, *** *p* < 0.001 or **** *p* < 0.0001.

**Figure 4 ijms-23-07316-f004:**
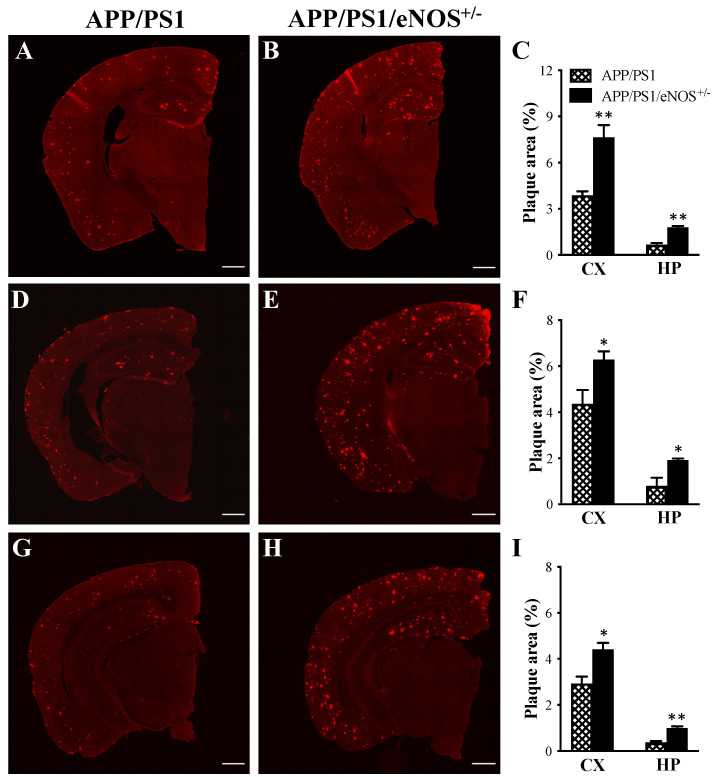
The representative 6E10 immunofluorescent coronal half-brain images of 8-months old male APP/PS1 (**A**,**D**,**G**) and APP/PS1/eNOS^+/−^ mice (**B**,**E**,**H**) at the levels of AP −1.70 (**A**,**B**), −2.30 (**D**,**E**) and −2.92 (**G**,**H**) mm (scale bar 600 µm). Mean (± SEM) percentage area of 6E10-positive staining (plaque area) in the cortex (CX) and hippocampus (HP) at the levels of AP −1.70 (**C**), −2.30 (**F**) and −2.92 (**I**) mm in APP/PS1 and APP/PS1/eNOS^+/−^ mice (*n* = 5–6/genotype). * indicates a significant difference between groups at * *p* < 0.05 or ** *p* < 0.01.

**Figure 5 ijms-23-07316-f005:**
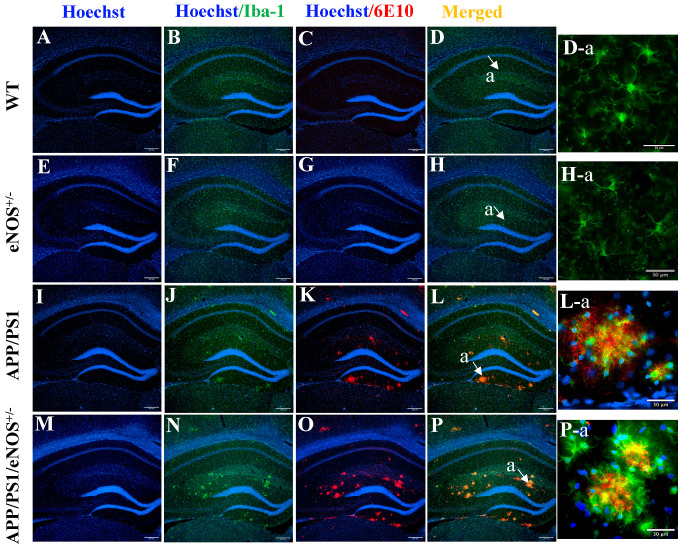
Representative immunofluorescent staining of Hoechst (blue; (**A**,**E**,**I**,**M**)), Hoechst/Iba-1 (green; (**B**,**F**,**J**,**N**)), Hoechst/6E10 (red; (**C**,**G**,**K**,**O**)) and merged Hoechst/Iba-1/6E10 (**D**,**H**,**L**,**P**) in the hippocampus of 8-months old male WT, eNOS^+/−^, APP/PS1 and APP/PS1/eNOS^+/−^ mice at 10× magnification (scale bar 200 μm). High magnification (40×) images of merged Hoechst/Iba-1/6E10 staining in the area labeled a in WT (**D-a**), eNOS^+/−^ (**H-a**), APP/PS1 (**L-a**) and APP/PS1/eNOS^+/−^ mice (**P-a**) (scale bar 30 μm).

**Figure 6 ijms-23-07316-f006:**
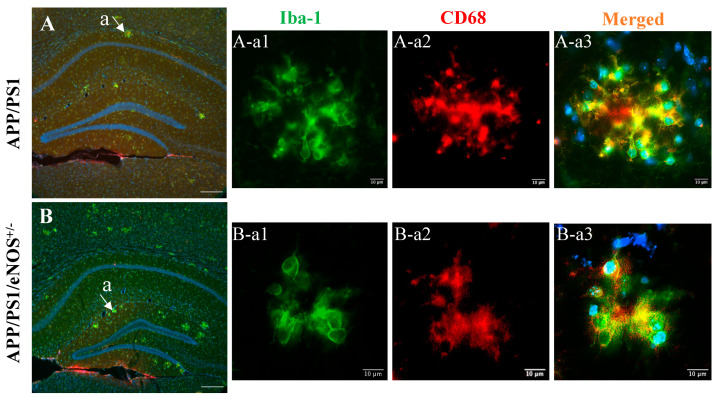
Representative merged images of Hoechst (blue), Iba-1 (green) and CD68 (red) in the hippocampus of 8-months old male APP/PS1 (**A**) and APP/PS1/eNOS^+/−^ (**B**) mice at 10× magnification (scale bar 200 μm). High magnification (40×) images of Iba-1, CD68 and merged Hoechst/Iba-1/CD68 staining in the area labeled a in APP/PS1 (**A-a1**,**A-a2**,**A-a3**) and APP/PS1/eNOS^+/−^ mice (**B-a1**,**B-a2**,**Ba-3**) (scale bar 10 μm).

**Figure 7 ijms-23-07316-f007:**
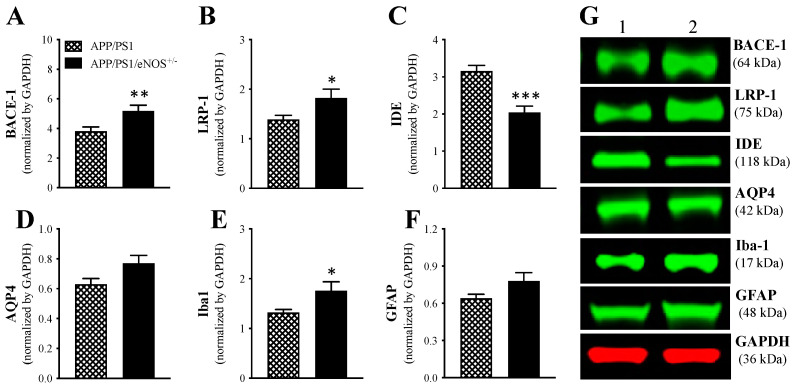
Mean (± SEM) protein expression of beta-site amyloid precursor protein cleaving enzyme 1 (BACE-1; (**A**)), low density lipoprotein receptor-related protein 1 (LRP-1; (**B**)), insulin-degrading enzyme (IDE, (**C**)), aquaporin-4 (AQP4, (**D**)), ionized calcium binding adaptor molecule 1 (Iba-1, (**E**)) and glial fibrillary acidic protein (GFAP, (**F**)) in the anterior cortex of 8-months old male APP/PS1 and APP/PS1/eNOS^+/−^ mice (*n* = 8–12/genotype). (**G**) representative western blots of BACE-1, LRP-1, IDE, AQP4, Iba-1 and GFAP, as well as housekeeping protein glyceraldehyde 3-phosphate dehydrogenase (GAPDH), in APP/PS1 (1) and APP/PS1/eNOS^+/−^ (2) mice. * indicates a significant difference between groups at * *p* < 0.05, ** *p* < 0.01 or *** *p* < 0.001.

**Figure 8 ijms-23-07316-f008:**
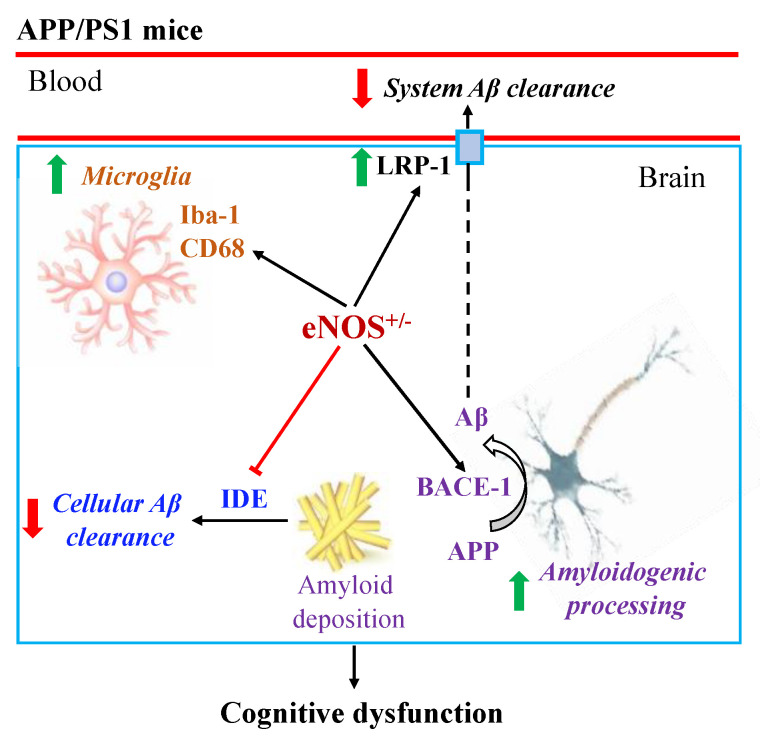
Schematic diagram illustrating how partial eNOS deficiency (eNOS^+/−^) aggravates cognitive dysfunction in APP/PS1 mice. The findings of the present study suggest that partial eNOS deficiency leads to augmented amyloidogenic processing of amyloid precursor protein (APP; hence increased amyloid beta (Aβ) production), reduced Aβ clearance and augmented microglial pathology in APP/PS1 mice, as evidenced by the upregulation of beta-site amyloid precursor protein cleaving enzyme-1 (BACE-1) and low density lipoprotein receptor-related protein-1 (LRP-1; speculatively a compensatory response), increased immunoreactivity of ionized calcium-binding adapter molecule 1 (Iba-1) and cluster of differentiation 68 (CD68), and the downregulation of insulin-degrading enzyme (IDE). These changes ultimately aggravate cognitive dysfunction in APP/PS1 mice.

## Data Availability

The data presented in this study are available from the corresponding author [P.L.] upon reasonable request.

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
