# Peer review of "Partial Endothelial Nitric Oxide Synthase Deficiency Exacerbates Cognitive Deficit and Amyloid Pathology in the APPswe/PS1ΔE9 Mouse Model of Alzheimer’s Disease"

_ijms, 2022, doi:10.3390/ijms23137316_

Round 1

Reviewer 1 Report

The work of Ahmed et al. it is really well written, the results appear convincing and the study design was well done. I also congratulate the authors for the quality of the images, it was a pleasure to see some beautiful immunofluorescence. All of this makes this paper certainly worth publishing in IJMS. I have only a few suggestions to implement in order to make the work more complete and more attractive to the reader.

Here are my comments:

- The authors investigate well the role of microglia in AD; it is known that astrocytosis is also involved in the neuroinflammatory process of Alzheimer's. Therefore I suggest evaluating GFAP by immunofluorescence technique, as was done for IBA-1.

- Figures 5 and 6 are missing a reference graph, in which the% of positive staining for each group and also the significance between the various experimental groups should be indicated.

- A brief mention of the experimental therapies in use and their limitations should be given at the beginning of the Discussion; this would explain why it is so important to find new research strategies.

- In my opinion, section 3 Discussion should be one big paragraph (combined with 3.1, 3.2 and 3.3)

-Figure 8 could be removed and used as Abstract graphical.

- Section 4.1: Did the procedures follow the ARRIVE guidelines? it should be indicated.

Author Response

“The work of Ahmed et al. it is really well written, the results appear convincing and the study design was well done. I also congratulate the authors for the quality of the images, it was a pleasure to see some beautiful immunofluorescence. All of this makes this paper certainly worth publishing in IJMS. I have only a few suggestions to implement in order to make the work more complete and more attractive to the reader.”

Response:

Thank you very much for such positive comments on our work!

“Here are my comments:

- The authors investigate well the role of microglia in AD; it is known that astrocytosis is also involved in the neuroinflammatory process of Alzheimer's. Therefore I suggest evaluating GFAP by immunofluorescence technique, as was done for IBA-1.”

Response:

Using western blot, we determined the Iba-1 and GFAP protein expression in the cortex between APP/PS1 and APP/PS1/eNOS+/- mice. The Iba-1 protein level was significantly increased in the APP/PS1/eNOS+/- group relative to the APP/PS1 group (Fig. 7E). However, there was no statistical significance between groups in terms of GFAP protein expression (Fig. 7F). We therefore only determined Iba-1 immunoreactive profile changes between groups.   

“- Figures 5 and 6 are missing a reference graph, in which the% of positive staining for each group and also the significance between the various experimental groups should be indicated.”

Response:

Immunofluorescent double labelling revealed intensely stained microglia forming clusters, which were colocalized with amyloid plaques in both the APP/PS1 and APP/PS1/eNOS+/- mice (Fig. 5). We quantified the Aβ load in the cortex and hippocampus in both groups (Fig. 4), however did not quantify the load of clustered microglia. Given the colocalization of clustered microglia and amyloid plaques and increased Aβ load in the APP/PS1/eNOS+/- group, partial eNOS deficiency appeared to augment microglial pathology in APP/PS1 mice. This conclusion was further supported by the western blot data showing significantly increased Iba-1 protein level in APP/PS1/eNOS+/- mice relative to APP/PS1 mice. The colocalization of clustered microglia labelled by Iba-1 and CD68 (Fig. 6) indicates a phagocytic phenotype of microglia associated with amyloid plaques, which we believe is more informative than area covered per se. We have discussed these points (see lines 424-438).

“- A brief mention of the experimental therapies in use and their limitations should be given at the beginning of the Discussion; this would explain why it is so important to find new research strategies.”

Response:

We have now added the following sentences at the beginning of the Discussion.

“Currently there are no effective therapies for AD, the most common dementia in the aged, despite decades of trials targeting Ab. Therefore, there is an urgent need to better understand other mechanisms underlying the disease process, to identify novel therapeutic targets and to create new animal models that better recapitulate the human disease state.” (see lines 310-314). It would unnecessarily extend the paper to discuss the experimental therapies in use and their limitations.

“- In my opinion, section 3 Discussion should be one big paragraph (combined with 3.1, 3.2 and 3.3)”

Response:

We have now removed the sub-headings as suggested (see pages 10-13).

“-Figure 8 could be removed and used as Abstract graphical.”

Response:

Figure 8 is now used as a graphical abstract as you suggested. We still like to keep Fig. 8 in the manuscript as a summary diagram. 

“- Section 4.1: Did the procedures follow the ARRIVE guidelines? it should be indicated.”

Response:

We have now indicated that our experimental procedures follow the ARRIVE guidelines (see lines 491 and 492).

Reviewer 2 Report

In this paper the authors generated mice with heterozygous deletion of endothelial nitric oxide synthase (eNOS) by crossing homozygous eNOS (-/-) mice with other mice carrying autosomal dominant familial Alzheimer disease (AD) genes that involve the amyloid precursor protein (APP mutations) and presenilin-1 (PS1) mutations that process APP (so-called mAPP/mPS1 mice). The resulting cross yielded a ~50-50 mixture of eNOS (+/-) and mAPP/mPS1/eNOS (+/-) mice.

The authors then showed that the mAPP/mPS1/eNOS(+/-) mice had increased b-amyloid plaque surface area and behavioral deficits compared to mAPP/mPS1 mice. This suggested that NOS deficit was altering b-amyloid metabolism, which the authors explored through Western blot of several proteins in brain extracts. They found that beta secretase (involved in APP processing to b-amyloid) protein was increased and insulin degrading enzyme (IDE, b-amyloid degrading enzyme) was decreased. The authors conclude that eNOS regulates b-amyloid levels in this particular animal model and may play a role in the much more common sporadic (non-familial) AD.

I have mixed emotions about this paper. On the one hand, the authors did a nice job in their experiments. There is nothing scientifically or methodologically wrong in their work. On the other hand, the amyloid cascade hypothesis is no longer accepted by many and to this reviewer is now considered a minority hypothesis for sporadic AD. Clearly at one time in the past it was prevalent and dominant; however, multiple failed clinical trials of b-amyloid reducing therapies have consistently met with failure, the recent FDA debacle notwithstanding.

So the relevance of the present study to sporadic AD is questionable at best, but still there may be insights, particularly about the varied roles of NO, that could flow from this paper. For that reason, I feel it is justified to publish this work. As an example, are the changes in b-secretase and IDE protein levels reflected in altered transcription of their genes? What about total (particularly "soluble") beta amyloid levels? Do mAPP/mPS1/eNOS (+/-) mice have increased brain cytokine levels coming from all those activated microglia?

I also have the following lesser concerns:

1. line 36. "the most" should be replaced a "a"; 2. several minor spelling corrections are needed; 3. the Discussion is too long and repetitive and can be shortened by at least one-half. 4. Was there any effect on survival to 8 mos of the eNOS (-/-) crossing?

Author Response

“In this paper the authors generated mice with heterozygous deletion of endothelial nitric oxide synthase (eNOS) by crossing homozygous eNOS (-/-) mice with other mice carrying autosomal dominant familial Alzheimer disease (AD) genes that involve the amyloid precursor protein (APP mutations) and presenilin-1 (PS1) mutations that process APP (so-called mAPP/mPS1 mice). The resulting cross yielded a ~50-50 mixture of eNOS (+/-) and mAPP/mPS1/eNOS (+/-) mice.

The authors then showed that the mAPP/mPS1/eNOS(+/-) mice had increased b-amyloid plaque surface area and behavioral deficits compared to mAPP/mPS1 mice. This suggested that NOS deficit was altering b-amyloid metabolism, which the authors explored through Western blot of several proteins in brain extracts. They found that beta secretase (involved in APP processing to b-amyloid) protein was increased and insulin degrading enzyme (IDE, b-amyloid degrading enzyme) was decreased. The authors conclude that eNOS regulates b-amyloid levels in this particular animal model and may play a role in the much more common sporadic (non-familial) AD.

I have mixed emotions about this paper. On the one hand, the authors did a nice job in their experiments. There is nothing scientifically or methodologically wrong in their work. On the other hand, the amyloid cascade hypothesis is no longer accepted by many and to this reviewer is now considered a minority hypothesis for sporadic AD. Clearly at one time in the past it was prevalent and dominant; however, multiple failed clinical trials of b-amyloid reducing therapies have consistently met with failure, the recent FDA debacle notwithstanding.

So the relevance of the present study to sporadic AD is questionable at best, but still there may be insights, particularly about the varied roles of NO, that could flow from this paper. For that reason, I feel it is justified to publish this work. As an example, are the changes in b-secretase and IDE protein levels reflected in altered transcription of their genes? What about total (particularly "soluble") beta amyloid levels? Do mAPP/mPS1/eNOS (+/-) mice have increased brain cytokine levels coming from all those activated microglia?”

Response:

We would like to thank the reviewer for your valuable and fair comments on the amyloid cascade hypothesis of AD, particularly its relevance to the late-onset sporadic AD. APP/PS1 mice have been considered as a model for the early stage of AD. Since there is growing evidence implicating cerebral endothelial dysfunction in the disease pathogenesis, APP/PS1/eNOS+/- mice offer a more clinically relevant model for early-stage AD with an element of endothelial dysfunction.  

In the present study, we determined how partial eNOS deficiency affected the protein (rather than mRNA) expression of BACE-1 and IDE in APP/PS1 mice, given the functional significance of proteins.

Regarding the total (particularly "soluble") beta amyloid levels, we have now added the soluble Aβ1-40 and Aβ1-42 data. There were no significant differences between APP/PS1 and APP/PS1/eNOS+/- mice (see lines 271-279 and 621-627).

This is a very good suggestion to look at the cytokine level difference between the two genotype groups. Unfortunately, the brain tissue samples were used for various neurochemical assays. The cytokine level issue will certainly be addressed in future research.    

“I also have the following lesser concerns:

  1. line 36. "the most" should be replaced a "a";”

Response:

We have now revised the sentence to “…., a commonly held view centres on the amyloid cascade hypothesis”.

“2. several minor spelling corrections are needed;”

Response:

We have now checked spelling and corrected typos.

“3. the Discussion is too long and repetitive and can be shortened by at least one-half.”

Response:

We have now revised the Discussion extensively as suggested.

“4. Was there any effect on survival to 8 mos of the eNOS (-/-) crossing?”

Response:

Based on our observations, partial eNOS deficiency did not affect the survival of APP/PS1 mice up to 8 months of age (see lines320-321). 

Round 2

Reviewer 1 Report

Nothing to add, the manuscript is acceptable in my opinion.